# In Vivo Quantification of Myocardial Amyloid Deposits in Patients with Suspected Transthyretin-Related Amyloidosis (ATTR)

**DOI:** 10.3390/jcm9113446

**Published:** 2020-10-27

**Authors:** Tim Wollenweber, Rene Rettl, Elisabeth Kretschmer-Chott, Sazan Rasul, Oana Kulterer, Eva Rainer, Markus Raidl, Michael P. Schaffarich, Sabrina Matschitsch, Michael Stadler, Tatjana Traub-Weidinger, Dietrich Beiztke, Christian Loewe, Franz Duca, Julia Mascherbauer, Diana Bonderman, Marcus Hacker

**Affiliations:** 1Division of Nuclear Medicine, Department of Biomedical Imaging and Image-Guided Therapy, Medical University of Vienna, 1090 Vienna, Austria; tim.wollenweber@meduniwien.ac.at (T.W.); elisabeth.kretschmer-chott@meduniwien.ac.at (E.K.-C.); sazan.rasul@meduniwien.ac.at (S.R.); oana.kulterer@meduniwien.ac.at (O.K.); eva.rainer@meduniwien.ac.at (E.R.); markus.raidl@akh-wien.at (M.R.); michael.schaffarich@meduniwien.ac.at (M.P.S.); sabrina.matschitsch@akh-wien.at (S.M.); michael.stadler@akh-wien.at (M.S.); tatjana.traub-weidinger@meduniwien.ac.at (T.T.-W.); 2Department of Internal Medicine II, Division of Cardiology, Medical University of Vienna, 1090 Vienna, Austria; rene.rettle@meduniwien.ac.at (R.R.); franz.duca@meduniwien.ac.at (F.D.); julia.mascherbauer@meduniwien.ac.at (J.M.); Diana.bondermann@meduniwien.ac.at (D.B.); 3Division of Cardiovascular and Interventional Radiology, Department of Biomedical Imaging and Image-Guided Therapy, Medical University of Vienna, 1090 Vienna, Austria; dietrich.beitzke@meduniwien.ac.at (D.B.); christian.loewe@meduniwien.ac.at (C.L.)

**Keywords:** amyloidosis, quantification, ATTR, SPECT/CT, SUV, bone scan

## Abstract

Background: Current diagnosis of Transthyretin-related Amyloidosis (ATTR) using bone scintigraphy is primarily based on visual scoring and semi-quantitative indices. With the introduction of new potential life-prolonging drugs for ATTR, a more precise quantification of myocardial amyloid burden is desirable for improved response prediction and therapy monitoring. Methods: At first, quantification experiments using an anthropomorphic thorax phantom were performed. Second, 32 patients underwent both planar whole body [^99m^Tc]- 3,3-Diphosphono-1,2-Propanodicarboxylic Acid (DPD)-scintigraphy and quantitative Single-Photon Emission Computed Tomography/Computed Tomography (SPECT/CT) of the thorax. SPECT/CT standardized myocardial uptake values SUVpeak and SUVpeak normalized to bone uptake (nSUVpeak) were determined. Results: Phantom measurements showed a strong linear relationship between the activity in the myocardial insert and the measured activity (r = 0.9998, *p* = 0.01), but the measured activity was systematically underestimated by approximately 30%. Receiver operating characteristics (ROC) analysis revealed a 100% sensitivity and specificity at a cut-off of 3.1 for SUVpeak for the differentiation of both patient groups. Conclusion: SUV quantification of ATTR amyloid burden is feasible using novel SPECT/CT technology. With a SUVpeak cut-off of 3.1, patients with Perugini grade 2 and 3 could be clearly separated from those with Perugini grade 0 and 1. Besides ATTR diagnostics, quantification of amyloid deposits could potentially be used for therapy monitoring and prognostication in patients with cardiac ATTR.

## 1. Introduction

Amyloidosis is a heterogeneous group of diseases characterized by the deposition of amyloid fibrils in the extracellular matrix of tissues and organs [1].

Deposition of amyloid fibrils in the myocardial extracellular matrix causes increased wall thickness leading to a deterioration of diastolic function [2]. The most common forms causing cardiac involvement are light-chain amyloidosis (AL), wild-type transthyretin (wtATTR) and hereditary transthyretin amyloidosis (mATTR) [3]. The latter is caused by point mutations in the TTR gene that lead to destabilization of transthyretin (TTR), a plasma protein responsible for the transport of thyroxine and vitamin A [4].

Recently, several drugs have been developed for the therapy of ATTR. These drugs are slowing down disease progression in patients with polyneuropathy [5,6] and reducing mortality in patients with cardiac ATTR [7,8]. With the introduction of these novel treatment options, both early diagnosis of cardiac amyloidosis as well as repetitive quantification of amyloid burden becomes increasingly important.

Modalities used for the diagnosis of cardiac amyloidosis are versatile ranging from electrocardiogram (ECG) and echography to cardiac magnetic resonance (CMR) imaging [9,10,11,12]. The latter visualizes the accumulation of amyloid in the extracellular matrix through an increase of the myocardial extracellular volume (ECV), which can also be quantified as previously described [13,14,15,16,17,18,19]. Despite a high sensitivity and specificity reported for various CMR and echocardiography approaches for the diagnosis of cardiac amyloidosis, their major limitation is the missing molecular information for the differentiation between different types of amyloidosis [16].

The current gold standard for amyloid detection and characterization is the histological examination of endomyocardial biopsy samples by staining with modified trichrome or specific amyloid antibodies. However, it is an invasive procedure frequently associated with the risk of sampling errors and the lack of availability of this technique.

Radiolabeled phosphonates like [^99m^Tc]- 3,3-Diphosphono-1,2-Propanodicarboxylic Acid (DPD) have a strong affinity for TTR amyloid fibril infiltrations in the heart [20]. The accuracy of DPD scintigraphy for the distinction of TTR-related and AL etiology has been shown to be very precise using genotyping and immunohistochemical analysis as a reference technique. Consequently, in the absence of monoclonal gammopathy, cardiac ATTR can be diagnosed without the need for additional histopathological examination by applying the quantification techniques in accordance with Perugini classification [21].

Perugini et al. developed a four-grade visual classification system of heart retention based on planar images 3 h after tracer injection (p.i.), where grade 0 is defined as absent cardiac uptake and normal bone uptake, grade 1 as mild cardiac uptake inferior to bone uptake, grade 2 as moderate cardiac uptake accompanied by attenuated bone uptake and grade 3 as strong cardiac uptake with mild/absent bone uptake.

Weaknesses of this scoring system are potential false positive results due to increased bloodpool activity of the radiopharmaceutical and a high reader dependency of this visual grading system. Therefore, although nuclear imaging is highly accurate for the diagnosis of cardiac ATTR, the visual Perugini grading is not suited for the evaluation of therapy monitoring or disease progression [22].

There have been attempts to use semi-quantitative indices like the heart to contralateral (H/CL) ratio [23,24] or the heart/whole-body ratio [25,26]. However, these indices may be affected by extra cardiac uptake in osteoblastic metastasis or lung uptake which may occur in ATTR patients. This potential limitation may be overcome by quantitative measurement of cardiac DPD uptake using novel Single-Photon Emission Computed Tomography/Computed Tomography (SPECT/CT) technology [27,28].

Previous studies have already correlated various quantitative SPECT parameters with Perugini scores [22] and found differences between patients with cardiac amyloidosis and healthy volunteers [29].

The present study aims to establish quantitative SPECT for the detection and therapy monitoring of cardiac ATTR. For this reason, we tested quantitative SPECT imaging in a series of phantom measurements before we investigated its feasibility in the clinical setting of suspected (and proven) cATTR.

## 2. Materials and Methods

### 2.1. Phantom Studies

Phantom experiments were performed on a dedicated SPECT/CT system (Symbia Intevo, Siemens Medical Solutions AG, Erlangen, Germany) equipped with a low-energy high-resolution collimator. Images were acquired in 180° configuration, 64 views, 20 s per view, 256 × 256 matrix and an energy window of 15% around the ^99m^Tc photopeak of 141 keV. Subsequent to the SPECT acquisition, a low-dose CT scan was acquired for attenuation correction (130 kV, 35 mAs, 256 × 256 matrix, step-and-shoot acquisition with body-contour). Images were reconstructed using the iterative xSPECT/CT QUANT algorithm (eight iterations, four subsets, 3.0 mm smoothing filter, and a 20 mm Gaussian filter).

An anthropomorphic thorax phantom with inserts to simulate lungs, left ventricular (LV) wall and LV chamber (PhantomTM, Data SpectrumCorporation, Hillsborough, NC, USA) was used. Myocardial wall activity concentration was ranged between 20.8 and 84.1 kBq/mL, while both the skeletal (43.8 kBq/mL) and the LV chamber background concentration (2 kBq/mL) of ^99m^Tc was kept stable. Bone tracer concentration was calculated based on the assumption, that after injection of 700 MBq, at the time point of image acquisition approximately 50 to 60% of the injected amount is fixed in the skeleton [30]. Calculated background concentration in the circulation was 0.5 kBq/mL (≈6% of the injected activity [30]), so that 2 kBq/mL were chosen taking also into account the high soft tissue uptake in Perugini grades 2 and 3.

### 2.2. Study Population

Thirty-two consecutive patients (9 women and 23 men; age 73 ± 11.2 years) with bioptically proven cATTR or suspected cardiac ATTR received a DPD total body bone scan with additional thorax SPECT/CT between 02/2019 and 10/2019. Patients with bioptically proven cATTR or suspicion due to transthoracic echocardiography or cMRI findings, hypertrophic cardiomyopathy, aortic valve or mitral clip implantation or known hereditary ATTR polyneuropathy were included in the study. All examinations were performed based on clinical indication. All patients gave written informed consent prior to imaging. Appropriate ethical approval was obtained by the Ethics Committee of the Medical University of Vienna (EK #769/2010).

### 2.3. Bone Scan

721 ± 25 MBq DPD were injected intravenously 2.5 h prior to whole body planar imaging on the same hybrid SPECT/CT system as the phantom experiments were conducted (Symbia Intevo, Siemens Medical Solutions AG, Erlangen, Germany). Directly after the planar scan, a SPECT/CT of the thorax was performed. Image acquisition and reconstruction was done using the same parameters as for the phantom experiments utilizing the xSPECT/CT QUANT technology, which uses a 3% National Institute of Standards and Technology (NIST) traceable precision source for standardization of uptake values across different cameras, dose calibrators, and facilities [31].

### 2.4. Image Analysis

Planar whole-body images were evaluated by two independent experienced observers. For visual grading Perugini scores were used as previously described [21]. Discrepancies were resolved by consensus.

Myocardial uptake on SPECT/CT images was determined using a commercially available software package (Hermes Hybrid 3D software, Hermes Medical Solutions, Stockholm, Sweden). Isocontour volumes of interest (VOIs) were generated using a 39% of the maximal activity threshold, which was developed with the phantom experiments. By applying this threshold myocardial uptake could be clearly separated from blood pool activity. From these VOIs, the SUVpeak was obtained. Bone uptake was calculated by placing a cubic 2.92 mL VOI in the center of an intact vertebral body of a thoracic spine. Using these values, SUVpeak was normalized to bone activity (nSUVpeak) as previously described [22]. Soft tissue SUVpeak was determined by placing a cubic 2.92 mL VOI into the subcutaneous fat of the left axillar region. As alternative parameters SUVpeak was weighted by multiplying with soft tissue uptake (wSUVpeak).

Heart to contralateral (H/CL) ratio was defined as the total counts in a region of interest (ROI) over the heart divided by counts in an identical size region of interest over the contralateral thorax, including soft tissue, ribs, and blood pool [23,24]. The heart/whole body ratio (H/WB) was assessed as previously described [26].

### 2.5. Genetic Analyses

Patients with imaging-confirmed ATTR underwent genetic testing. This group consisted of 7 patients with mATTR (1xThr80Ala, 4xHis108Arg; 1xVal113Leu, 1xVal50Met) and 14 patients with wtATTR. Myocardial SUVpeak, nSUVpeak and wSUVpeak as well as soft tissue and bone SUVpeak values were compared for these genetic subgroups.

### 2.6. CMR

In a subgroup of 15 patients CMR was available. CMR was performed using a 1.5 T scanner (Siemens Avanto Fit) and standardized protocols including late gadolinium enhancement and T1 mapping for calculation of the ECV [32,33].

### 2.7. Statistical Analysis

Statistical analysis was performed using MedCalc v19.1 (Ostend, Belgium). Linear regression analysis was performed to describe the relationship between variables. The Kolmogorov–Smirnov test was used to check for normal distribution. Normally distributed data were expressed as mean ± standard deviation. Comparisons of different Perugini grade groups with normal distribution were performed using unpaired Student’s *t*-test. If normal distribution was rejected, comparisons between groups with different Perugini grades were performed using Wilcoxon–Mann–Whitney U test. ROC analysis was performed to determine reference intervals and clinical decision limits. The Bonferroni correction was used in the case of multiple hypothesis tests. *p* < 0.05 was considered as statistically significant.

## 3. Results

### 3.1. Phantom Experiments

First, a threshold of the maximum activity for the automatic contouring of the volume of interest (VOI) was developed in phantom experiments. Here, a threshold of 39 percent of the maximum activity led to a volume of the VOI of approximately 155 mL in all three measurements, which was in good agreement with the heart insert volume of 155 mL. Applying this threshold resulted in a clear linear relationship between the applied and the measured activity concentration, which was confirmed by linear regression analysis (r = 0.9998, *p* = 0.01; Figure 1). The *Bland*–*Altman* plot shows that the ratio between applied and measured activities were spread around a mean of 1.5. This indicates that the activity was underestimated about one third.

### 3.2. Study Population

In total, 22 of the 32 patients had confirmed diagnosis of ATTR (69%). In 14 of those patients, the diagnosis of cardiac ATTR was confirmed by Perugini scores of 2 and 3 (in absence of a monoclonal protein) and in eight patients by biopsy of the left ventricular myocardium. Figure 2 shows representative examples of the Perugini grading.

15/32 patients had additional cardiac MRI scans from which the ECV was determined. The mean time between MRI and bone can was 98 ± 193 days. The characteristics of the patients are summarized in Table 1.

### 3.3. Bone Scans

Based on planar images 4/32 patients (13%) were rated as Perugini grade 0 and 5/32 (16%) as Perugini grade 1, while 9 (28%) and 14 (44%) patients were rated as grade 2 and 3, respectively.

In three-dimensional SPECT/CT, myocardial SUVpeak, nSUVpeak as well as wSUVpeak were significantly increased in the group of patients with Perugini scores 2 and 3 compared to the patients with Perugini scores 0 and 1 (13.3 ± 5.1 vs. 1.7 ± 0.7; *p* < 0.0001; 2.5 ± 1.1 vs. 0.3 ± 0.2; and 1.1 ± 0.6 vs. 13.3 ± 6.8 respectively; *p* < 0.0001; see also Appendix A). The majority of patients with Perugini scores 2 and 3 showed a strong regional uptake in the septal wall and less in the apical region. Slight uptake was also found in the right ventricular myocardium and in the arterial tree in some patients. 3/5 patients with Perugini score 1 showed no significant myocardial uptake in SPECT/CT (SUVpeak values for these patients were 1.8, 1.2 and 1.7). The remaining 2 patients with Perugini grade 1 showed slightly increased uptake (SUVpeak 2.3 and 3.1), but were not definitly diagnosed at the current time point.

ROC analysis revealed 100% sensitivity and specificity at a SUVpeak cut-off of 3.1 to differentiate between patients with Perugini 2 and 3 vs. 0 and 1. Interestingly, SUVpeak was significantly increased in patients with Perugini score 2 compared to Perugini score 3 (16.5 ± 4.7 vs. 11.1 ± 4.4; *p* = 0.011); a difference that, however, disappeared after background correction (nSUVpeak 2.4 ± 1.2 vs. 2.7 ± 1.0; *p* = 0.531 Figure 3b) and after weighting with soft tissue uptake (wSUVpeak 13.2.0 ± 8.2 vs. 13.6 ± 6.0; *p* = 0.95 Figure 3c). As shown in Appendix A linear regression analysis showed a significant correlation of both myocardial SUVpeak, SUVpeak and wSUVpeak with Perugini scores.

In addition, there was a strong correlation of both semi-quantitative parameters H/CL and H/WB with Perugini grades (see Appendix A
Appendix A).

No correlation was found for both myocardial SUVpeak and nSUVpeak with ECV (r = 0.07; *p* = 0.81 and r = 0.20; *p* = 0.48; Figure 4a,b). However, there was a significant correlation of wSUVpeak (r = 0.61; *p* = 0.015 Figure 4c). Furthermore, ECV showed a strong linear relationship with Perugini scores (r = 0.76; *p* = 0.0009) as shown in Appendix A there was no correlation of H/CL and H/WB with ECV ((r = 0.005; *p* = 0.99 and r = 0.31; *p* = 0.26, respectively; Appendix A).

### 3.4. Genetic Correlations

Myocardial SUVpeak was significantly increased in patients with wtATTR compared to patients with mATTR (*p* = 0.003), while soft tissue SUVpeak was significantly increased in patients with mATTR (*p* = 0.005). For bone SUVpeak no significant difference was observed between patients with mATTR and wtATTR (*p* = 0.36). Myocardial nSUVpeak was significantly increased in patients with wtATTR (*p* = 0.01), while myocardial wSUVpeak and ECV were significantly increased in patients with mATTR (*p* = 0.0001 and *p* = 0.008, respectively). Moreover, Perugini grades were significantly increased in patients with mATTR (*p* = 0.02; Figure 5).

## 4. Discussion

The present study demonstrates the feasibility of novel SPECT/CT techniques to quantify myocardial ATTR amyloid deposits in vivo. In phantom experiments, a linear relationship between applied and measured activity was demonstrated, even if the measured activity concentration was systematically underestimated by approximately 30%. The latter finding is probably caused by partial volume effects, taking into account the width of the wall of the heart insert of approximately 1.0 cm and the Full Width at Half Maximum (FWHM) of the used camera system of 7.5 mm [34]. In a set of 32 consecutive patients with suspected cardiac ATTR, in vivo SUVpeak values for myocardial and bone uptake were in the same range as reported in other studies [22,29,35,36], indicating reliability and reproducibility of this novel quantification technique. Even if one considers the small sample size and the high pretest probability of disease the 100% sensitivity and specificity allowed the establishment of a cut-off value of 3.1 for SUVpeak to clearly distinguish patients with Perugini scores 2 and 3 from those with Perugini scores 0 and 1, which is necessary to separate patients with high probability from those with low probability for cardiac ATTR [37]. The cutoff is in good agreement with previous works for DPD [22] and HDP [29]. Using this threshold, it is possible to differentiate between patients with and without high probability cardiac ATTR without additional scoring of planar images with Perugini scores.

Interestingly, a large overlap between patients with Perugini scores 2 and 3 was demonstrated for quantitative SPECT/CT values, an effect which has been previously observed [22,36,38]. This observation is also in good agreement with the work of Hutt et al. [39] who found that cardiac amyloid burden, determined by equilibrium contrast cardiac magnetic resonance imaging, was similar between patients with Perugini grade 2 and 3. Additionally, Perugini et al. [21] found a similar spectrum of LV masses determined by echocardiography irrespective of the visual scintigraphic scoring results.

A likely explanation for the large overlap of myocardial SUVpeak values in patients with Perugini grades 2 and 3 may be that the high soft tissue uptake in Perugini grade 3 patients may lead to a competition for radiotracer uptake between myocardial, bone and soft tissue. Compatible with this explanation is the fact that in the present study myocardial SUVpeak values were clearly increased in Perugini grade 2 compared to Perugini grade 3 patients, a difference which nearly vanished for the background corrected nSUVpeak and soft tissue weighted wSUVpeak values. It should be pointed out here that this is in disagreement with the fact that ECV showed a trend to be increased in patients with Perugini grade 3 compared to Perugini grade 2 (*p* = 0.052). In our study, there was no significant correlation between cardiac SUVpeak and ECV measured by MRI, which is contrast to Scully et al. [36] who found a good correlation between cardiac SUVpeak and ECV measured by CT. However, it should be pointed out that they excluded Perugini score 3 from their analysis to avoid that quantification of cardiac amyloid burden is confounded by competition for the radiotracer from surrounding soft tissue. Furthermore, besides the fact that there was a linear correlation found between Perugini grades and ECV, which was also observed in a previous study (20), there was also a linear correlation found between wSUVpeak and ECV. This indicates that a weighting with soft tissue background activity (wSUVpeak) should be applied for SPECT/CT quantification of myocardial amyloid deposits. This could be the utilized for in therapy monotoring of the former mentioned novel drugs for the treatment of cardiac ATTR.

The latter statement is also supported by the fact that there was a strong correlation with the semi-quantitative parameters H/CL an H/WB, which have shown high sensitivity and specificity for differentiation of patients with cardiac ATTR from those with cardiac AL and may also be of diagnostic and prognostic importance. Additionally the correlation was more significant for H/WB, which seems to be the more accurate parameter [25], than H/CL.

Several recent studies have also shown the value of positron emission (PET) imaging with amyloid tracers such as [C11]-Pittsburgh compound B (PIB) [40], [F^18^]^-^ florbetapir [41], [F^18^]^-^ florbetaben [42,43] or [F^18^]^-^ flutemetamol [44] for the diagnosis of cardiac amyloidosis. Furthermore, it has been demonstrated that myocardial tracer retention (for florbetaben) correlated well with morphologic and functional parameters, as measured by CMR and echo, and with treatment response [42]. Although PET is expected to have more accurate quantification abilities than SPECT, studies revealed that SPECT might be superior compared to PET for the distinction of the underlying subtype of cardiac amyloidosis, as PET uptake intensity of cardiac AL and ATTR largely overlaps [42,43,44,45]. Another advantage of conventional bone scintigraphy is its broad availability compared to amyloid PET.

We also compared SUV quantification and Perugini scores for patient subgroups with wtATTR and mATTR. While patients with wtATTR showed increased myocardial uptake (SUVpeak and nSUVpeak) with predominantly lower and atypical soft tissue patterns involving the gluteal, shoulder, chest, and abdominal wall region, patients with mATTR showed a generally increased soft tissue uptake, with compared to wtATTR lower myocardial tracer uptake, again supporting the assumption that there is a competition for tracer uptake between both compartments. When SUVpeak was weighted with soft tissue background activity (wSUVpeak), then wSUVpeak was significantly increased in patients with mATTR compared to wtATTR, which is which corresponds to the fact that ECV was higher in patients with mATTR again indicating that wSUVpeak should be applied for quantification of myocardial amyloid deposits.

The decreased soft tissue uptake in patients with wtATTR may be the reason that the sensitivity of abdominal fat aspiration seems to be lower in patients wtATTR compared with patients mATTR [46].

As previously reported, myocardial DPD uptake in SPECT/CT images was more prominent in septal and basal regions than in lateral and apical regions [38], which is in good agreement with the characteristic relative apical sparing of longitudinal strain in echocardiography and indicates that apical amyloid deposits may be only apparent in advanced cardiac ATTR.

This study has several limitations. The first limitation is the relatively small sample size of subjects, which is not unusual for such a rare disease. Secondly, at the current time point subjects with Perugini grade 1 scans were not further evaluated for cardiac AL. Thirdly, ECV was only measured in a subset of patients, which could have influenced the observed associations. Fourthly, endomyocardial biopsy as gold standard for confirmation of ATTR was not available in all patients. However, patients without endomyocardial biopsy were diagnosed based on confirmed Perugini scores of 2 and 3 and the absence of a monoclonal protein [37].

In conclusion, patients with high and low probability for cardiac ATTR can be clearly separated using a novel SPECT/CT technique with a SUVpeak cut-off of 3.1. Furthermore, the good correlation of soft tissue weighted SUVpeak with ECV determined by cMRI indicates that myocardial amyloid deposits can be reliably quantified using this parameter. However, this conclusion must be validated in subsequent studies by comparison with the histological processing of endomyocardial biopsies. Nevertheless, the results of this study indicate that SPECT/CT SUV quantification could potentially be applied for therapy monitoring to provide more insights in therapy induced changes of amyloid deposits.

## Figures and Tables

**Figure 1 jcm-09-03446-f001:**
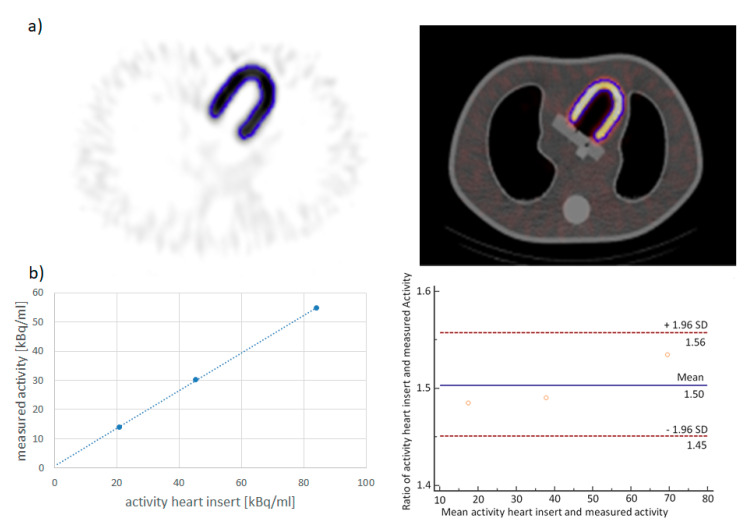
Phantom experiments: (**a**) The anthropomorphic thorax phantom with an activity of 20.8 kBq/mL in the heart insert and an activity of 2.0 kBq/mL in the background. A threshold of 39 percent of the maximum activity met the myocardial wall of the heart insert very well; (**b**) Relationship between the activity in the myocardial wall insert and the measured activity. The best fit straight line fits the data points very well (r = 0.9998, *p* = 0.01). Bland–Altman plot demonstrates a spread of the ratio between applied and measured activity arround a mean of 1.5.

**Figure 2 jcm-09-03446-f002:**
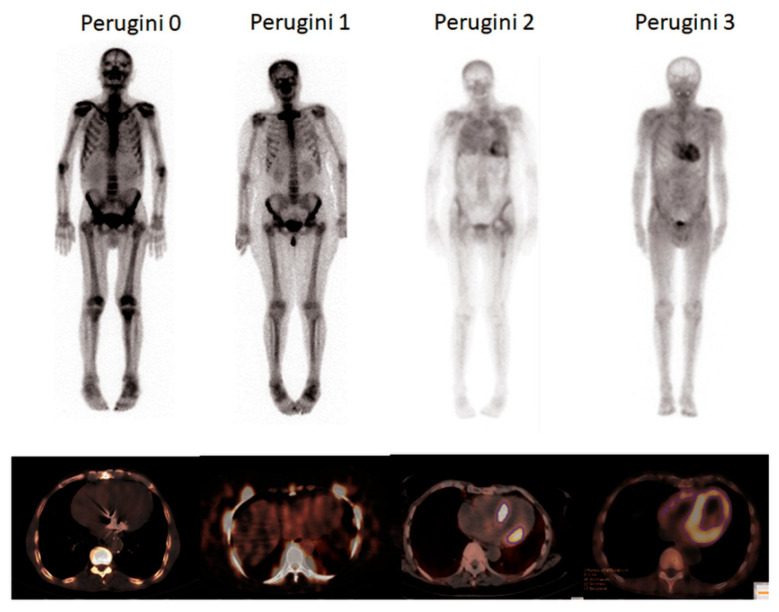
Patient examples for all four Perugini scores: This figure shows an example of a patient with every Perugini score.

**Figure 3 jcm-09-03446-f003:**
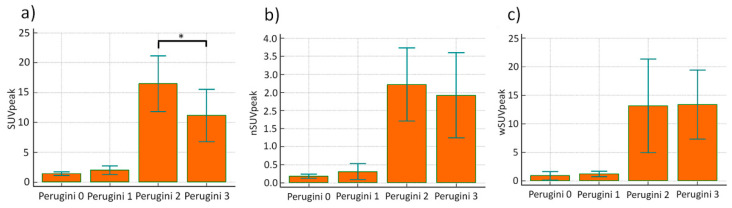
Comparison of SUVpeak, nSuvpeak and wSUVpeak values for Perugini grades 0–3: (**a**) SUVpeak was significantly increased in patients with Perugini grade 2 compared to 3 (* *p* = 0.0011), whereby the standard deviation of around 50% has to be taken into consideration, so that the significance might disappear with higher patient numbers; (**b**) nSUVpeak showed no significant differences between both values (*p* = 0.531); (**c**) wSUVpeak also showed no significant differences between Perugini grade 2 and 3 (*p* = 0.95).

**Figure 4 jcm-09-03446-f004:**
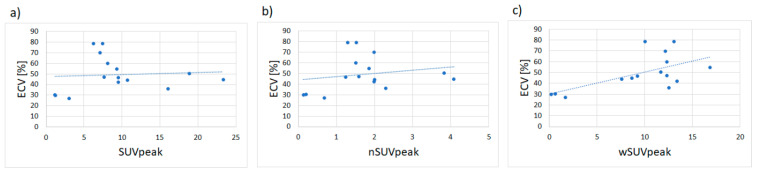
Correlation of SUVpeak, nSUVpeak and wSUVpeak with ECV: (**a**) and (**b**) SUVpeak, nSUVpeak showed no correlation with ECV (r = 0.07; *p* = 0.81 and r = 0.20; *p* = 0.48); (**c**) while wSUVpeak showed a good correlation with ECV (r = 0.61; *p* = 0.015).

**Figure 5 jcm-09-03446-f005:**
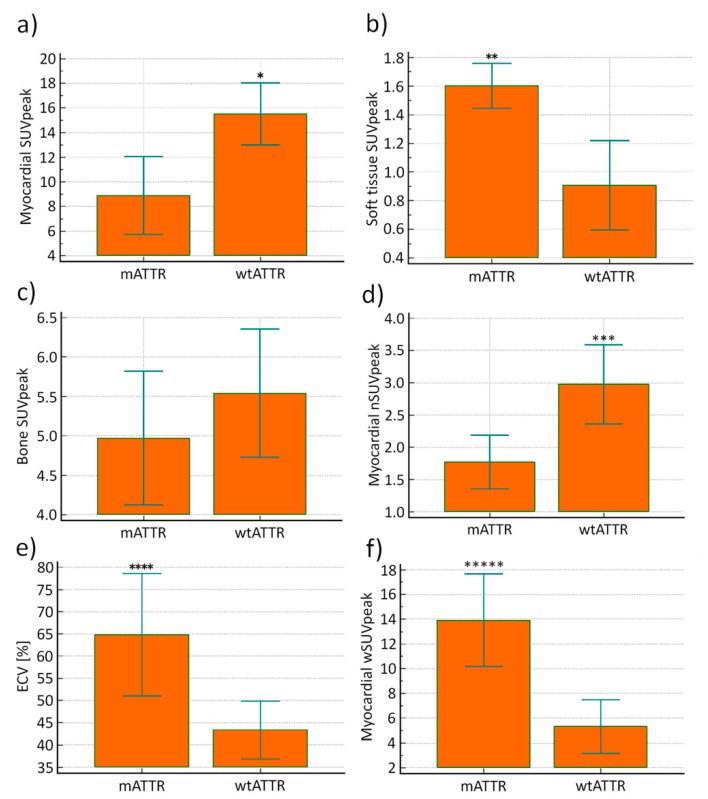
Genetic correlations. (**a**) Myocardial SUVpeak was significantly increased in patients with wtATTR compared to patients with mATTR (* *p* = 0.003). (**b**) In contrast soft tissue SUVpeak was significantly increased in patients with mATTR (** *p* = 0.005). (**c**) Furthermore, there was no significant difference found for bone SUVpeak between patients with wtATTR and mATTR (*p* = 0.36). (**d**) Myocardial nSUVpeak was significantly increased in patients with wtATTR (*** *p* = 0.01). (**e**) ECV (***** *p* = 0.008) and (**f**) Myocardial wSUVpeak (***** *p* = 0.0001) were significantly increased in patients with mATTR, respectively.

**Table 1 jcm-09-03446-t001:** Characteristics of the study cohort: The characteristics of the 32 patients included in the study with age, gender, myocardial and bone SUVpeak, ECV (if available), Perugini score and genetic mutation.

Patient no.	Age	Gender	SUVpeakMyocardium	ECV [%]	Perguini Score	Suvpeak Bone	Genetic Mutation
1	60	F	8.1	60	3	5.4	His108Arg
2	66	M	1.0	-	0	9.5	
3	88	F	10.7	43.9	2	5.4	wtATTR
4	79	M	2.3	-	1	8.0	
5	64	M	9.3	54.5	3	5.0	Thr80Ala
6	62	F	7.5	78.7	3	5.8	His108Arg
7	77	M	16.1	35.8	2	7	wtATTR
8	78	F	22.4	-	2	5.8	wtATTR
9	71	F	9.6	42.1	3	4.8	wtATTR
10	84	M	18.9	50.3	2	4.9	wtATTR
11	67	M	7.1	69.9	3	3.6	His108Arg
12	74	M	23.4	44.8	2	5.7	wtATTR
13	80	M	14.4	-	3	3.9	wtATTR
14	54	F	1.8	-	1	12.7	
15	87	M	1.2	30.1	1	5.8	
16	82	M	10.5	-	3	4.1	wtATTR
17	63	F	1.7	-	1	7.5	
18	78	M	8.4	-	3	7.6	wtATTR
19	86	M	9.5	46.6	2	7.6	unknown
20	79	M	18.8	-	3	3.6	wtATTR
21	48	M	6.2	78.8	3	4.1	His108Arg
22	70	F	16.4	-	3	6.2	Val113Leu
23	75	M	7.6	47	3	4.8	Val50Met
24	90	M	16.6	-	2	5.5	wtATTR
25	78	M	19.1	-	3	6.6	wtATTR
26	79	M	12.9	-	3	3.4	wtATTR
27	74	M	14.3	-	2	6.8	wtATTR
28	61	F	1.3	-	0	6.2	
29	79	M	1.9	-	0	7.3	
30	77	M	16.8	-	2	8.2	wtATTR
31	80	M	3.1	27	1	4.5	
32	45	M	1.2	29.76	0	9.3

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
