# Peer review of "In Vivo Quantification of Myocardial Amyloid Deposits in Patients with Suspected Transthyretin-Related Amyloidosis (ATTR)"

_jcm, 2020, doi:10.3390/jcm9113446_

Round 1

Reviewer 1 Report

It seems to me an interesting topic and very well addressed, with interesting results that are summarized in a good way, in my capacity as reviewer I think I do not have comments for specific changes in the document

Reviewer 2 Report

In the submitted paper,Wollenweber et al. demonstrates the feasibility of novel SPECT/CT techniques to quantify myocardial ATTR amyloid deposits in-vivo. The manuscript is well presented and well written. Methods and results sounds right to me.

Major recommendation :

A discussion including the recent results using positron emission tomography in the assessment of cardiac amyloidosis is warranted to add more context. References such as : Kircher M, Ihne S, Brumberg J, Morbach C, Knop S, Kortüm KM, Störk S, Buck AK, Reiter T, Bauer WR, Lapa C. Detection of cardiac amyloidosis with 18F-Florbetaben-PET/CT in comparison to echocardiography, cardiac MRI and DPD-scintigraphy. Eur J Nucl Med Mol Imaging. 2019 Jul;46(7):1407-1416. doi: 10.1007/s00259-019-04290-y. Epub 2019 Feb 23. PMID: 30798427. should be discussed  

Minor corrections :

A few synthax errors:

- Page 4, Line 142 : "und" instead of "and" 

- Page 10, Line 274 : "MRT" instead of MRI I guess

- Page 10, Line 282: "therapymonotoring"

Reviewer 3 Report

The search for novel techniques for distinguish the patients with high and low probability for cardiac ATTR is of a great interest. The more precise quantification of myocardial amyloid burden is desirable for therapy monitoring and better prognosis. The present study has sought the unique novel SPECT/CT technique with a SUVpeak cut-off and using a soft tissue weighted SUVpeak. The Authors propose a scenario whereby ATTR  patients may be distinguish between Perugini grade 2 and 3 without endomyocardial biopsy.

  1. The Introduction should be more reader friendly. There are a lot of the clinical jargon (some examples are added in the specific comments, but not all of them). In addition, there is not clear why gold standards techniques used for ATTR qualification are not suitable. Thus, Authors should expand their minds.

  1. The Perugini score should be more precise explained. Also the weak points of this qualification should be mentioned.

  1. The study has got some limitations such as low number of patients. However, it is a rare disease. The biggest concern is that not all patients with ATTR had a endomyocardial biopsy. For this reason the conclusion of the study is risky.

 Specific comments

L19: In the abstract the abbreviations should be explained at first usage.

L43: amyloidosis (mATTR) amyloidosis – please correct that.

L46: double space – please edit that.

L47: please use abbreviation ATTR because it means Transthyretin-related Amyoidosis so usage two times amyloidosis is useless.

L63: once again ATTR amyloidosis

L64: the clinical jargon is used. It should be histological examination/or histopathological examination

L64: maybe Perugini classification? This classification should be explained.

L93: double space

L204/Figure 3 – SD is really high (almost 50%) so the significance can disappear.

Round 2

Reviewer 2 Report

To my opinion, the second version of the manuscript seems adequate for publication.

Reviewer 3 Report

Dear Authors, the quality of the manuscript now is developed. The paper has been improved.